# Detection of Methicillin Susceptible and Resistant *Staphylococcus aureus* Nasal Carriage and Its Antibiotic Sensitivity among Basic and Clinical Years Medical Students

**DOI:** 10.3390/healthcare8020161

**Published:** 2020-06-06

**Authors:** Hamed Alzoubi, Mohammad Al Madadha, Asma’a Al-Mnayyis, Muayad Azzam, Amira Aldawoud, Danah Hwaiti, Mohammad Tarbiah, Maha Abu Ajamieh, Mamoon Qatamin

**Affiliations:** 1Department of Microbiology and Immunology, Faculty of Medicine, Mu’tah University, Karak 61710, Jordan; 2Faculty of Medicine, Mutah University, P.O. Box 7, Karak 61710, Jordan; 3Department of Pathology and Microbiology and Forensic Medicine, Faculty of Medicine, University of Jordan, Amman 11924, Jordan; m.madadha@ju.edu.jo; 4Department of Clinical Sciences, College of Medicine, Yarmouk University, Shafiq Irshaidat St, Irbid 221136, Jordan; asmaa.mnayyis@yu.edu.jo; 5Faculty of Medicine, University of Jordan, Amman 11924, Jordan; may0160150@ju.edu.jo (M.A.); amy0162292@ju.edu.jo (A.A.); dan0160212@ju.edu.jo (D.H.); mhm0160888@ju.edu.jo (M.T.); MHA0162308@ju.edu.jo (M.A.A.); Mam0162338@ju.edu.jo (M.Q.)

**Keywords:** methicillin resistant *Staphylococcus aureus* (MRSA), methicillin susceptible *Staphylococcus aureus* (MSSA), nasal carriage, medical students, antibiotic susceptibility

## Abstract

*Background:* Healthcare workers (HCWs) and medical students can be asymptomatic carriers in transmitting methicillin resistant and susceptible *Staphylococcus aureus* (MRSA and MSSA). Studying epidemiological and antibiotic susceptibility data is necessary to limit the spread of infections, help with treatment and understand the transmission dynamics of MSSA and MRSA. Our study assessed the rate of MSSA and MRSA nasal carriage and its antibiogram among medical students in basic and clinical years at the University of Jordan. *Methods:* A total of 210 nasal swabs were randomly collected from participants. MSSA and MRSA were identified by culture, biochemical and other phenotypical analysis methods. Antibiotic susceptibility was determined by the disc diffusion method. *Results:* The nasal carriage of MSSA was 6.6% and 11.4% and that of MRSA was 1.9% and 2.8% among basic and clinical years, respectively. There was no significant difference for the nasal carriage of MSSA and MRSA among basic and clinical year students (*p* value ≥ 0.05). MSSA resistance ranged between 25% and 33% for trimethoprim-sulfamethoxazole, tetracycline and ciprofloxacin. For MRSA, the highest resistance was to trimethoprim-sulfamethoxazole and tetracycline (67% to 100%), followed by gentamicin and ciprofloxacin (33% to 67%), in all participants in the study. *Conclusion:* The difference in the carriage rates of MSSA and MRSA among basic and clinical students was statistically insignificant. The continuous awareness and implementation of infection control procedures and guided patient contact are recommended. The results might also suggest that healthcare workers could be victims in the cycle of MRSA nasal carriage, a theory that needs further study.

## 1. Introduction

*Staphylococcus aureus* (*S. aureus*) is considered one of the most common commensal bacteria that is usually found in the anterior nares of healthy individuals, but it is also a major pathogen that has been associated with serious community and nosocomial acquired infections, such as skin and soft tissue infections, pneumonia and bloodstream infections, which are associated with increased hospital mortality and morbidity [1,2]. The ability to develop resistance to various classes of antibiotics has brought more clinical attention to *S. aureus*. Soon after the introduction of methicillin, resistance emerged in certain strains of *S. aureus,* and these strains became famously described as methicillin resistant *Staphylococcus aureus* (MRSA), which was first isolated in the UK in the early 1960s [3]. Since then, the prevalence of MRSA has increased and it has been found that MRSA proportion constituted between 20% and 80% of all *S. aureus* isolates in different countries [2].

MRSA developed its resistance by acquiring the *mec* A gene which codes for an altered penicillin binding protein (PBP2a) with a reduced affinity to beta-lactam antibiotics, causing resistance to virtually all beta-lactam antibiotics [4,5]. In addition, MRSA usually shows resistance to a wide range of other commonly prescribed anti-staphylococcal antibiotics, which limits therapeutic options, making it necessary to use specialized antibiotics such as glycopeptides and linezolid to treat MRSA associated infections [6]. This emerging resistance has led to increased mortality and morbidity rates, length of hospital stays and costs of treatment [2].

Factors such as inadequate infection control practices, antibiotic misuse, chronic co-morbidities, recurrent hospitalization and repeated exposure to healthcare settings, as well as certain host factors such as immunosuppression, have all been suggested to increase the risk of colonization by MRSA and its spread [7,8]. In many countries, including Jordan, it was revealed that as much as 50% of all invasive *S. aureus* infections were caused by MRSA [9]. This mandates that more studies should be performed in order to further investigate the transmission dynamics, risk factors, epidemiology and practices that may contribute to such high rates of MRSA spread. Previous studies suggested that colonized healthcare personnel act as a reservoir that may serve as a vector in the transmission cycle of MRSA, which increases the likelihood of MRSA spread within the hospital and into the community [10,11]. Previous studies in Jordan showed that nasal carriage of MRSA among healthcare workers ranged from 5.8% to 8.7% [12,13]. Taking into consideration the resistance pattern of MRSA, the ease of spread and the fatal outcomes of infections caused by MRSA [10], it is therefore necessary to identify the potential risk factors, prevalence and antibiotic resistance pattern of MRSA in medical students as part of the healthcare forces, given that medical students may have little or no awareness of infection prevention and control practices, which can contribute significantly to the spread of nosocomial infections. Therefore, the screening of the medical students would be an essential part of a multifactorial approach in order to understand the spread behavior of MSSA and MRSA, to protect health personnel, patients and the community and to encourage the application of better infection prevention precautions and increase awareness [10,14].

The aim of this study was to assess the carriage rates of methicillin susceptible *Staphylococcus aureus* (MSSA) and MRSA among medical students, from the basic first three years of medical study, who had not been not exposed to hospital practices, and compare the results to those of students in their last three clinical years of study (hospital based training years) and to assess the antibiotic susceptibility patterns of the isolated bacteria.

## 2. Materials and Methods

### 2.1. Study Design, Population and Data Collection

This cross-sectional study was conducted at the Faculty of Medicine in the University of Jordan, Amman, Jordan over a 6 month period in 2018/2019. Based on the results of the previous study that was conducted in Jordan, which showed that the prevalence of positive MRSA amongst healthcare staff is around 6% [12], the sample size required for this study was 100 samples [15]. Therefore, a total of 210 nasal swabs were randomly collected from medical students in their basic years (105 samples) and the same number from clinical year students. The age of the basic year students ranged from 19 to 21 years with a total of 50 females and 55 males. For clinical years, the age ranged from 22 to 24 years with 56 females and 49 males. Written questionnaires that were signed and considered as informed consent to participate were collected from each participant before nasal specimen collection. Using the questionnaire that was filled in by all students, the data that were collected included answers about age, sex, study stage, recent hospital admission or antibiotic exposure over the last 3 months and having a first degree relative who was a healthcare worker (HCW). All participants with a recent hospital admission or antibiotic consumption or with a first degree relative who was an HCW were excluded from the study. The study was approved by the Faculty of Medicine Ethics Committee, approval number 201833, since it respected the legal disposition that applied to it as reported in the supplementary materials.

### 2.2. Swab Cultures, Bacterial Strains and Antimicrobial Susceptibility Testing

Sampling for each participant was performed by rotating a sterile cotton swab in the vestibule of both anterior nares, as previously described by Wen–Tsung et al. [16]. Cotton swabs were spread immediately on mannitol salt agar (MSA) (MSA; BBL Microbiology Systems, Becton Dickinson, Company, Sparks, MD, USA). The collected samples were transported within 1–2 h at a temperature of 4–8 °C to the Microbiology Laboratory. Plates were incubated aerobically at 35 ± 1 °C and examined for growth after 24–48 h. Each distinctive morphotype of mannitol-fermenting yellow colony was subcultured in a nutrient agar (BBL Microbiology Systems, Becton Dickinson, Company, Sparks, MD, USA) and incubated aerobically at 37 °C in a humidified incubator. Colonies growing on the nutrient agar were identified as *S. aureus* by their typical colony morphology, Gram’s staining, anaerobic utilization of glucose and mannitol, catalase production and tube coagulase test. Screening for methicillin resistance was done using a 30 μg/mL cefoxitin disc in Mueller–Hinton agar, according to Clinical and Laboratory Standard Institute (CLSI) guidelines [17]. Antibiotic susceptibility tests were performed using Kirby Bauer’s disc diffusion method, according to the performance standards of the European Committee on Antimicrobial Susceptibility Testing for fusidic acid and according to the CLSI guidelines for the remaining antibiotics [17,18]. *S. aureus* ATCC 25923 and ATCC 33591 were used as control strains for phenotypic identification. The panel of antibiotics that was tested was selected based on those that were recommended by CLSI or were commonly used locally in the empirical treatment of *S. aureus* infection. Susceptibility testing was done for the following antibiotics: trimethoprim-sulfamethoxazole, ciprofloxacin, tetracycline, gentamicin, linezolid, mupirocin, fusidic acid, rifampicin and vancomycin. For MRSA, a multi-drug resistant (MDR) isolate is defined in this study as one which is resistant to 3 other different classes of antibiotics. All MRSA isolates were frozen at −70 °C for additional testing.

### 2.3. Statistical Analysis

The statistical analysis was performed with STATA13 (Stata Corp. 2013. Stata Statistical Software: Release 13. StataCorp LP, College Station, TX, USA) to evaluate the significance of results. A *p* value ≤ 0.05 was considered as significant using the chi-square test.

## 3. Results

### 3.1. The Rate of MSSA and MRSA Detection among the Study Sample

The rate of MSSA and MRSA detection is shown in Table 1. The MSSA nasal carriage in basic year medical students was about 6.6% (7/105), and in clinical year students, it was 11.4% (12/105). On one hand, there was no significant difference between the nasal carriage of MSSA in each group (*p* = 0.22). On the other hand, the MRSA nasal carriage was about 1.9% in basic year medical students and was 2.8% in clinical year students. There was no significant difference between the two groups (*p* = 0.65).

### 3.2. Antibiotic Susceptibility of MRSA and MSSA Isolates from Basic and Clinical Year Students

The antibiogram of all isolates is shown in Table 2. The highest level of resistance among the seven MSSA isolates from basic students was 28% (2/7 isolates) against tetracycline. All MSSA isolates were susceptible to the remaining antibiotic.

Among the clinical students, 33% of MSSA isolates were only resistant to trimethoprim-sulfamethoxazole (4/12 isolates) and 25% to ciprofloxacin (3/12 isolates). No multidrug resistant MSSA was detected.

Contrastingly, 100% (2/2 isolates) of MRSA isolates from basic students were resistant to trimethoprim-sulfamethoxazole and tetracycline, while 50% (1/2 isolates) were resistant to ciprofloxacin and gentamicin. Overall, one MRSA isolate was resistant to all four classes of antibiotics and defined as an MDR isolate. MRSA isolates from clinical students showed a 67% (2/3 isolates) resistance to trimethoprim-sulfamethoxazole, tetracycline and gentamicin, while 33% (1/3 isolates) were resistant to ciprofloxacin. Overall, two MRSA isolates were identified as MDR isolates. None of the MRSA isolates were resistant to linezolid, mupirocin, fusidic acid, rifampicin or vancomycin.

## 4. Discussion

*Staphylococcus aureus* is a major cause of serious infection and colonization, with both MSSA and MRSA representing a major risk for developing subsequent infections [19]. The risk factors for nasal carriage are still not fully understood and studies are continuing to define such factors [20].

HCWs (including training medical students) are on the interface between the community and patients. Their role might put them at increased risk of being carriers or victims of the cross transmission of *S. aureus* (MSSA or MRSA) [10]. Studying the carriage rate of both MSSA and MRSA and their antibiotic sensitivities among medical students represents an important tool that may prove useful for infection prevention measures, treatment and understanding of the dynamics of exposure to healthcare settings. The teaching classes of the basic students in our study were delivered at the university campus, which was separate from the clinical students, who received their classes and training bedside at the nearby teaching hospital. We therefore speculated that carriage rates were to be different among these two groups. Our study has shown that the number of MSSA and MRSA isolates detected among basic year medical students was lower than in those isolated from clinical students; however, the difference was statistically insignificant. Therefore, we recommend the periodic screening of students alongside future studies containing larger sample sizes, which may aid in the further exploration of such issues.

The MSSA nasal carriages were 6.6% and 11.4% for basic and clinical students, respectively. This is lower than what was recently found by Al-Tamimi et al. (2018), where the MSSA carriage rate was 19.3% among medical students at Hashemite University in Jordan [21]. Such variance could be explained by the difference in the study population and the presence of other variable risk factors among the randomly selected cohorts in the Al-Tamimi study (such as recent antibiotic consumption, presence of a relative who is a healthcare worker or a recent hospital admission); all these factors were excluded in our study population. Additionally, we included a commitment to infection prevention precautions by students through education prior to conducting the sampling, which is another important factor that should be included in future studies. However, the MRSA carriage rates in our study were 1.9% and 2.8% for basic and clinical students, respectively, which is comparable to the overall rate of 2.4% found by Al-Tamimi et al. (2018). Similar studies on medical students in nearby countries (Saudia Arabia and Iran) showed MRSA carriage rates of 6.7% to 13%, respectively [22,23], which are much higher than those found in our study.

Antibiotic resistance among the MSSA isolates in our study showed overall highly sensitive *S. aureus* strains, with resistance ranging between 25% and 33% for trimethoprim, sulfamethoxazole, tetracycline and ciprofloxacin. The other antibiotics used in the study showed a 100% sensitivity. Among MRSA, the highest resistance was for trimethoprim sulfamethoxazole and tetracycline with a range of 67%–100%, followed by gentamicin and ciprofloxacin with a range of 33% to 67%. Such resistance patterns are nearly comparable to what was previously found in Jordan among medical students and healthcare workers [13,21]. The resistance pattern of MSSA for trimethoprim-sulfamethoxazole, tetracycline and gentamicin found in our study was lower than that found previously in other countries [24] and was slightly higher (15.7% vs. 11.6%) for ciprofloxacin. Low resistance levels for the remaining antibiotics, such as vancomycin, mupirocin and fusidic acid, are in agreement with other studies which have been conducted worldwide [24,25].

In our study, for MRSA, lower resistance rates for trimethoprim-sulfamethoxazole, tetracycline, ciprofloxacin and gentamicin were found compared to other studies in different countries, while the resistance to fusidic acid and mupirocin was comparable [24,25,26]. Variability in resistance to some antibiotics is expected and might be explained by factors such as the presence of variable phylogenetic strains, difference in factors present in each study population and different exclusion criteria used by each study, such as recent hospital admission or antibiotic use. Both these factors may increase the risk of selecting strains with more resistant tendencies.

## 5. Conclusions

In conclusion, the findings of the current study showed an overall low nasal carriage rate of MSSA and MRSA among medical students at the University of Jordan, but the difference was not statistically significant. This should be regularly monitored and attempts to lower such rates would be highly desirable. This can be achieved via multidisciplinary efforts aiming to implement strict infection control procedures and guided patient contact. Additionally, finding a low rate of MRSA carriage among medical students when compared to other studies which showed a higher rate of MRSA carriage among professional HCWs [12,13], might support the idea that HCWs can be victims of MRSA nasal carriage and later act as vectors in transmission. However, further studies are still required to support such hypotheses by screening similar age groups in non-healthcare related environments, such as factories, offices or gyms, and comparing the carriage rates of MSSA and MRSA could be useful and is highly recommended in future studies.

## Figures and Tables

**Table 1 healthcare-08-00161-t001:** number and percentage of MSSA and MRSA among basic and clinical years medical students.

Pathogen	Population		*p*-Value
	Basic Years (Sample = 105) (Number, %)	Clinical Years (Sample = 105) (Number, %)	
MSSA	7 (6.6%)	12 (11.4%)	0.22
MRSA	2 (1.9%)	3 (2.8%)	0.65

MSSA, methicillin susceptible *Staphylococcus aureus*; MRSA, methicillin resistant *Staphylococcus aureus*; +ve, positive; significant *p* value ≤ 0.05.

**Table 2 healthcare-08-00161-t002:** Antibiotic resistance pattern of MSSA and MRSA isolates from basic and clinical year medical students.

Population	Pathogen/No.	Antibiotic Resistance (Number, %)								
		SXT	Cip	Tet	G	LZD	Mup	FD	Rif	Van
**Basic years**	**MSSA/7**	0;0	0;0	2;28	0;0	0;0	0;0	0;0	0;0	0;0
	**MRSA/2**	2;100	1;50	2;100	1;50	0;0	0;0	0;0	0;0	0;0
**Clinical years**	**MSSA/12**	4;33	3;25	0;0	0;0	0;0	0;0	0;0	0;0	0;0
	**MRSA/3**	2;67	1;33	2;67	2;67	0;0	0;0	0;0	0;0	0;0

SXT, trimethoprim-sulfamethoxazole; Cip, ciprofloxacin; Tet, tetracycline; G, gentamicin; LZD, linezolid; Mup, mupirocin; FD, fusidic acid; Rif, rifampicin; Van, vancomycin.

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
