# Peer review of "Detection of Methicillin Susceptible and Resistant Staphylococcus aureus Nasal Carriage and Its Antibiotic Sensitivity among Basic and Clinical Years Medical Students"

_healthcare, 2020, doi:10.3390/healthcare8020161_

Round 1

Reviewer 1 Report

An interesting study, I think the paper could be accepted in its current form. However, it would have been useful to put in a control MRSA strain for comparison in table 2. Also is the spectrum of resistance in the students consistent with that found worldwide and in that age/socio-economic group?

I don't think you can state that basic-years students have less carriage rates than the other student population as the p value was above 0.5. You state that that is because they don't have hospital access, but surely they are in the same environment as the other students and the transmission could happen then? You would have expected the presence of MRSA to be significantly higher in the students in their later years?

I think a concurrent study looking at a group of people of the same age would be of interest and make the paper much stronger? I would suggest profiling a group of 18-24's who work together in a workplace not associated with an university? Gym, factory, office etc? Then a statistical analysis of these populations would give a better understanding of nocosomial vs non-nocosomial spread. 

Author Response

Response to reviewer 1:

First paragraph:

1.Note regarding MRSA control:

Our response: It is mentioned in lines 112-113 that S. aureus ATCC 25923 and ATCC 33591 were used as controls, it is further clarified this time by adding that it is used for phenotypical identification line 112. This was also briefly rephrased to respond to the reviewer valuable comment and clarify that. Additionally, the expected resistant pattern of any detected MRSA strain is usually unpredictable as it depends on many variable risk factors that study population were exposed to such as immunosupression, antibiotic misuse and abuse and recent or recurrent hospital admissions. In case of our study and every other study, the resistance pattern of isolated strains might be cited for comparison in other future studies.

2.Note regarding if spectrum of resistance in the students consistent with that found worldwide and in that socio-economic group?

Our response: we answered the suggestion as required by a paragraph that was added at the end of discussion in lines 204-216 and supported by new 3 references (references 25-27).

Paragraph 2 – only one point raised by the reviewer:

Our response: we already mentioned in our submitted manuscript, results section  in lines 129 and 131 that the MSSA and MRSA carriage rate difference between basic and clinical students was not statistically significant and repeated that in discussion in line 173. This time, we clarified that in discussion again by clarifying that basic years students do have their lectures in university halls completely separate from hospital environment, while clinical years students have their lectures and seminars in hospital halls.  We also added a note that the number of MRSA and MSSA was low in basic students rather than referring to that as  the carriage rate and we stressed again that observation was not statistically significant (Lines 171 and 173)

Paragraph 3 – only one point raised by the reviewer:

The idea is significant, and it advised performing a study in similar age group but in a different environment such as a gym or a factory.

Our response: we highly appreciate this idea and it has now been suggested in last paragraph in conclusion section lines 228-230  as a future work for coming studies. Such a study will represent a new completely different idea that will represent a continuation/new field of study.

Reviewer 2 Report

The manuscript by Hamed Alzoubi on detection of MRSA and MSSA in medical students (basic and clinical year’s stages) and the evaluation of antibiotic susceptibility is interesting and addresses the important topic for infectious diseases: the duality commensal/pathogen. Nevertheless, the manuscript has major issues that must be corrected/improved.

General comment: The manuscript will benefit from editing by an English native speaker. It will improve clarity.

Specific comments:

  • Check affiliations (specially number 3) and use it in a consistent way in the author's line
  • Review all the abstract content. Specific note for line 29: was indifferently low. What do you mean with this? If there is no statically significant difference, this should be stated clearly.
  • Line 33 and 34: Keywords. Instead of sensitive use susceptible (this applies to all document) and introduce the abreviature after the name. E g: methicillin resistant S. aureus (MRSA)       
  • Line 36-72 Review the introduction to improve clarity
  • Line 74. Provide proof of approval  by the ethical committee for this study
  • Lines 103-104 check if the font used for trimethroprim-sulfamethoxazole is correct
  • Lines 105-107  Revise the definition of MRSA
  • Lines 120-121 consider rearranging the table. It is not easy to read. I suggest that a first column with pathogen (n/%) and a second column with population that will be divided in basic year and clinical year students. The p values could be introduced only in the text.
  • Lines 139-142. I suggest a similar rearrangement of this table. Introduce an additional column with population and introducing here the year in which the medical students are. This is not a pathogen and should not be in that column.
  • Line 143-174 Improve the discussion. In material and methods the authors wrote "data about age, sex, study stage....(lines 82-83)" were collected but these data was neither presented nor discussed. The authors only mentioned in lines 158 to 160 a few factors that were used to exclude participants from the study. Please explain this point and either discuss the collected data or specify the admission/ exclusion criteria for participants selection.                                                                                                                 

Conclusion (starting line 175). Revise this section and correct the font colour in line 182.

Author Response

Response to reviewer 2:

The reviewer general comment: which was regarding improving English language in general

Our response: We read through the whole MS again and improved/edited the language again when required to increase clarity; done by the second author Dr Al Madadha who is an American citizen working as an assistant professor, and Author Abu Ajamieh who is a volunteer content editor AskHakeem.

The reviewer specific comments:

  • The affiliation number 3 4 was edited and now it is in author’s line (line 11, 13)
  • The abstract contents reviewed and slightly edited. Also, line 29 mentioned by the reviewer has been modified from indifferently low To the difference was statistically insignificant as suggested by the reviewer (line 32).
  • Reviewer comments on line 33 and 34: sensitive changed to susceptible in the whole MS, and the abbreviations placed after the names as suggested (lines 2,17,36, 135). Also abbreviation placed after the name lines (36 and 37)
  • Lines 36-72: introduction was reviewed, and clarity improved as part in response to general comment about English language, editing can be seen in the revised MS.
  • Lines 74: a copy of ethical approval proof has been provided in our response.
  • Lines 103-104: font of trimethoprim -sulfamethoxazole has been corrected line 115 now.
  • Lines 105-107: definition of MRSA MDR reviewed, and we added: (in this study) (classes of) in line 116.
  • Lines 120-121 table 1: we rearranged table 1 as per reviewer suggestions lines (132-136)
  • Lines 139-142 Table 2: we rearranged table 2 as per reviewer suggestions (lines 152-155)
  • Lines 143-174 discussion edited and improved, also we clarify the reviewer query regarding exclusion criteria by editing discussion in lines 189-192 to specifically highlight our exclusion factors

In the same point the reviewer query regarding age and sex of participants in methods lines 82-83 as stated by the reviewer we responded by clarifying the age ranges of basic and clinical students and their gender as seen in methods in lines 90-92

  • Conclusion line 175: the whole conclusion (lines 219-230) reviewed and edited as required  And The font colour has been corrected (line 225).

Round 2

Reviewer 2 Report

The manuscript was significantly improved but the following issues should be addressed:

Lines 6-7. Follow the journal instructions

Firstname Lastname 1, Firstname Lastname 2 and Firstname Lastname 2,*

Line 29.”… tetracycline (67%-100%),…” Use “to” instead of “-“as has been done for the other antimicrobial agents.

“… in all year.” Consider all participants in the study.

Lines 35-36. Revise keywords

Line 40. “but can also a major pathogen that has….” The verb is missing. Like this, the sentence has no meaning.

Lines 39-42 “Staphylococcus aureus (S. aureus) is considered one of the common commensal bacteria that is usually found in the anterior nares of healthy individuals, but can also a major pathogen that has been associated with serious community and nosocomial acquired infections with increasing 41 morbidity and mortality rates [1].”

and

Lines 46-48 “MRSA; which has been first isolated in the UK in early 1960s, still represents a challenging 46 infection both as a nosocomial and as a community acquired pathogen, which is associated with 47 increasing morbidity and mortality rates [2,3].”

The text is very similar only the references are different. The same applies to the explanation of the methicillin resistance. The first 2 paragraphs should be rearranged in order to report the prevalence of S. aureus infections and the major challenges of MRSA infections.

Line 55 “certain host factors” provide examples of host factors.

Line 78. Section “2.1. Study design, population and data collection”. In this section should be written that the study was approved by an ethic committee since it respects the legal dispositions that apply to it referring the latest clearly.

Line 103. Replace sensitivity by susceptibility.

Line 105. CLSI instead of Clinical Laboratory Standards Institute

Line 125, Table 1. What is the meaning of “MSSA +ve” and “MRSA+ve”? Why the “+ve” in the table and its absence in the legend. Please explain or correct.

Lines 131-132. “All MSSA isolates were 131 susceptible to the remaining antibiotic profile.” The word “profile” should be removed. Consider, …to the remaining antibiotics or antibiotics tested.

Lines 141-142. “None of the MRSA isolates in all students were resistant to linezolid, mupirocin, fusidic acid, rifampicin or vancomycin.” Consider removing in all students.

Table 2: Either use (number, %) or (number, %) in all table sections.

Line 149. Representing instead of represents

Lines 156 -158. “Our study has showed that the number of MSSA and MRSA isolates detected among basic-years medical students was lower than in those isolated from clinical students, however, the difference was statistically insignificant.” From here, we conclude that there is no statistically significant difference in MSSA/MRSA carriage in the two groups.

After explaining the hypothesis, the conclusion is repeated in lines 161-162 “in our study the difference was shown without 161 statistical significance.”

Please organize the text (lines 156-162) in order to avoid repeating.

In lines 168-169 after referring the study by Al-Tamimi the authors wrote “(such as recent antibiotic consumption, 168 presence of a relative who is a healthcare worker or a recent hospital admission) all these factors 169 were excluded in our study population.” But in lines 87-89 the authors wrote: “Using the questionnaire that was filled by all students, data collected included data about age, sex, study stage, recent hospital admission, and antibiotic exposure over the last 3 months.” Why was these data collected? The authors should either relate the results to the data collected in the questionnaire or state that individuals with certain characteristics were excluded from the study (e.g. taking antibiotics over the last 3 months, etc).

This question was asked in the last report and was not conveniently addressed.

Lines 194-195: “(notably such as recent hospital admission or antibiotic use, both these 194 factors may increase the risk of selecting strains with more resistance tendency.” There is an open bracket please revise.

Author Response

The manuscript was significantly improved but the following issues should be addressed:

We would like to thank the reviewer for the enormous efforts and precious comments that will surely enrich the MS. Please see our responses below each comment – red colored.

Lines 6-7. Follow the journal instructions

Firstname Lastname 1, Firstname Lastname 2 and Firstname Lastname 2,*

Response: Corrected as per journal instructions

Line 29.”… tetracycline (67%-100%),…” Use “to” instead of “-“as has been done for the other antimicrobial agents.

Response: we used ‘to’

“… in all year.” Consider all participants in the study.

Response: we replaced in all years to in all participants in the study

Lines 35-36. Revise keywords

Response: keywords have been reviewed

Line 40. “but can also a major pathogen that has….” The verb is missing. Like this, the sentence has no meaning.

Response: we added verb ‘be’

Lines 39-42 “Staphylococcus aureus (S. aureus) is considered one of the common commensal bacteria that is usually found in the anterior nares of healthy individuals, but can also a major pathogen that has been associated with serious community and nosocomial acquired infections with increasing 41 morbidity and mortality rates [1].”

and

Lines 46-48 “MRSA; which has been first isolated in the UK in early 1960s, still represents a challenging 46 infection both as a nosocomial and as a community acquired pathogen, which is associated with 47 increasing morbidity and mortality rates [2,3].”

The text is very similar only the references are different. The same applies to the explanation of the methicillin resistance. The first 2 paragraphs should be rearranged in order to report the prevalence of S. aureus infections and the major challenges of MRSA infections.

Response: the paragraphs were rewritten to avoid repetition and to highlights the specific challenges of MRSA such as increasing the mortality and morbidity rates, length of stay and costs of treatment. Also the paragraphs highlighted the high prevalence of  MRSA among the isolated S. aureus isolates in different countries. The order of references 2 and 3 was corrected accordingly.

Line 55 “certain host factors” provide examples of host factors.

Response: we added ‘immunosuppression; as an example of an important host factor, the reader usually knows that many reasons might cause immunosuppression such as congenital defects, use of certain immunosuppression medications and certain infections such as HIV or any other illness that might lower immunity such as diabetes. It is known that immunity is necessary to fight infections. We summerised all of these by adding immunosuppression as the main example.

Line 78. Section “2.1. Study design, population and data collection”. In this section should be written that the study was approved by an ethic committee since it respects the legal dispositions that apply to it referring the latest clearly.

Response: sentence added as requested in last part of section 2.1.

Line 103. Replace sensitivity by susceptibility.

Response: replaced

Line 105. CLSI instead of Clinical Laboratory Standards Institute

Response: replaced

Line 125, Table 1. What is the meaning of “MSSA +ve” and “MRSA+ve”? Why the “+ve” in the table and its absence in the legend. Please explain or correct.

Response: we added +ve, positive. MSSA and MRSA abbreviations already exist in the legend

Lines 131-132. “All MSSA isolates were 131 susceptible to the remaining antibiotic profile.” The word “profile” should be removed. Consider, …to the remaining antibiotics or antibiotics tested.

Response: we removed “profile”

Lines 141-142. “None of the MRSA isolates in all students were resistant to linezolid, mupirocin, fusidic acid, rifampicin or vancomycin.” Consider removing in all students.

Response: removed

Table 2: Either use (number, %) or (number, %) in all table sections.

 Response: we changed all to (number, %) 

Line 149. Representing instead of represents

Response: we changed represents to representing

Lines 156 -158. “Our study has showed that the number of MSSA and MRSA isolates detected among basic-years medical students was lower than in those isolated from clinical students, however, the difference was statistically insignificant.” From here, we conclude that there is no statistically significant difference in MSSA/MRSA carriage in the two groups.

After explaining the hypothesis, the conclusion is repeated in lines 161-162 “in our study the difference was shown without 161 statistical significance.”

Please organize the text (lines 156-162) in order to avoid repeating.

Response: The lines were organized in a way repetition is not existing anymore

In lines 168-169 after referring the study by Al-Tamimi the authors wrote “(such as recent antibiotic consumption, 168 presence of a relative who is a healthcare worker or a recent hospital admission) all these factors 169 were excluded in our study population.” But in lines 87-89 the authors wrote: “Using the questionnaire that was filled by all students, data collected included data about age, sex, study stage, recent hospital admission, and antibiotic exposure over the last 3 months.” Why was these data collected? The authors should either relate the results to the data collected in the questionnaire or state that individuals with certain characteristics were excluded from the study (e.g. taking antibiotics over the last 3 months, etc).

Response: The questionnaire we provided with the supplementary files included specific questions about recent hospital admissions (last 3 months) and antibiotic consumption and having a first degree relative who is a HCW, all these were used as exclusion criteria and we referred to that in discussion when we said that ‘’All these factors were  excluded in our study population’’. However, the reviewer is right and to avoid confusing the reader we clarified this point by adding a sentence in methods in section 2.1 saying that:

All participants with a recent hospital admission or antibiotic consumption or having a first degree relative who is a HCW were excluded from the study.

This question was asked in the last report and was not conveniently addressed.

Please see response above

Lines 194-195: “(notably such as recent hospital admission or antibiotic use, both these 194 factors may increase the risk of selecting strains with more resistance tendency.” There is an open bracket please revise.

Response: bracket removed

Round 3

Reviewer 2 Report

Consider the following:

Lines 35-37: Keywords: Methicillin Resistant Staphylococcus aureus (MRSA); Methicillin Susceptible Staphylococcus aureus  (MSSA); Nasal carriage; Medical students; Antibiotic susceptibility 

Table 1. The isolates in the two population were either MSSA or MRSA not MSSA positive or MRSA positive. Either correct or explain.

Line 166: instead of has showed consider has shown.

I think the manuscript is acceptable as a communication.

Author Response

Comments and Suggestions for Authors

Consider the following:

Lines 35-37: Keywords: Methicillin Resistant Staphylococcus aureus (MRSA); Methicillin Susceptible Staphylococcus aureus  (MSSA); Nasal carriage; Medical students; Antibiotic susceptibility

Response: we did as suggested by putting abbreviations in brackets and we used susceptibility  

Table 1. The isolates in the two population were either MSSA or MRSA not MSSA positive or MRSA positive. Either correct or explain.

Response: we did as suggested, we removed positive to make scientific

Line 166: instead of has showed consider has shown.

Response: corrected to has shown

Many thanks for the significant notes.